# Redefining Precision Management of r/r Large B-Cell Lymphoma: Novel Antibodies Take on CART and BMT in the Quest for Future Treatment Strategies

**DOI:** 10.3390/cells12141858

**Published:** 2023-07-14

**Authors:** Reyad Dada

**Affiliations:** 1King Faisal Specialist Hospital and Research Centre, Jeddah 21499, Saudi Arabia; r-dada@web.de; Tel.: +966-2-6677777 (ext. 64065); Fax: +966-2-6677777 (ext. 64030); 2College of Medicine, Al-Faisal University, Riyadh 11533, Saudi Arabia

**Keywords:** large B-cell lymphoma, LBL, CART, bispecific antibodies, antibody–drug conjugate

## Abstract

The treatment paradigms for patients with relapsed large B-cell lymphoma are expanding. Chimeric antigen receptor technology (CAR-T) has revolutionized the management of these patients. Novel bispecific antibodies and antibody–drug conjugates, used as chemotherapy-free single agents or in combination with other novel therapeutics, have been quickly introduced into the real-world setting. With such a paradigm shift, patients have an improved chance of better outcomes with unpredictable complete remission rates. Additionally, the excellent tolerance of new antibodies targeting B-cell lymphomas is another motivation to broaden its use in relapsed and refractory patients. With the increasing number of approved therapy approaches, future research needs to focus on optimizing the sequence and developing new combination strategies for these antibodies, both among themselves and with other agents. Clinical, pathological, and genetic risk profiling can assist in identifying which patients are most likely to benefit from these costly therapeutic options. However, new combinations may lead to new side effects, which we must learn to deal with. This review provides a comprehensive overview of the current state of research on several innovative antibodies for the precision management of large B-cell lymphoma. It explores various treatment strategies, such as CAR-T vs. ASCT, naked antibodies, antibody–drug conjugates, bispecific antibodies, and bispecific T-cell engagers, as well as discussing the challenges and future perspectives of novel treatment strategies. We also delve into resistance mechanisms and factors that may affect decision making. Moreover, each section provides a detailed analysis of the available literature and ongoing clinical trials.

## 1. Introduction

In the recently updated “5th Edition of the World Health Organization Classification of Hematolymphoid Tumors”, large B-cell lymphoma (LBCL) has been included as a group of lymphomas that encompasses 18 different entities [1]. It represents the most common types of non-Hodgkin lymphoma (NHL) with aggressive behavior and distinct histopathological and clinical features, with diffuse large B-cell lymphoma (DLBCL) being the most common subtype. The annual incidence in the United States of America during 1992–2001 was around 7 cases per 100,000 persons [2], and this is projected to increase by 4% by 2025. The median age at diagnosis is 66 years with 52% of newly diagnosed patients having advanced stages (stage III and IV). The 5-year relative survival rate has improved from 61% (2002–2007) to 64% (2008–2013) [3]. This slight numerical improvement is statistically significant (*p* < 0.0001) and is related to several contributing factors, such as fewer patients being diagnosed with human immunodeficiency virus and DLBCL, increased usage of intensive chemotherapy regimens, and improved supportive care, including advances in the management of infectious complication [3]. However, despite all of our achievements, we are still left with 40% of patients who will experience primary refractory disease or eventually relapse, with only around 50% of these patients qualifying for autologous stem cell transplantation (ASCT) [4]. More recently, the treatment landscape of LBCL has been enriched by the endorsement of chimeric antigen receptor T-cell therapy (CAR-T) in earlier lines. Its approval has been expanded to include patients with primary refractory disease or relapses (r/r) within 12 months.

Treatment decisions for patients with r/r LBCL have become more complex and should take into account several factors, including the duration of remission, individual patient characteristics, and available treatment options (Figure 1). This review will focus on novel antibodies and discuss their outcomes in relevant clinical studies. The discussion will also encompass the safety profiles and provide information on ongoing studies. Furthermore, the review will explore challenges and potential future prospects associated with these antibodies.

## 2. First Salvage Treatment Options in r/r LBCL

DLBCL represents by far the most common subtype of LBCL, and approximately 30–40% of patients with DLBCL will experience a relapse, with an additional 10% having primary refractory disease. The data for other LBCL subtypes are not significantly different [5,6]. Relapsed and especially primary refractory disease behaves aggressively, and untreated patients may die within 3–4 months [7]. Historically, there have been various second-line treatment options, ranging from well-tolerated to more intensive regimens. However, the outcome of patients with primary refractory and early-relapsed LBCL treated with salvage conventional chemotherapy remains poor, with only 7% of patients achieving complete remission (CR) [8]. Several novel oral agents have been explored with none achieving approval. Ibrutinib, an oral covalent Bruton’s tyrosine kinase inhibitor, led in a phase I/II trial to 14/38 and only 1/20 responses in patients with the activated B-cell (ABC) and germinal center B-cell (GCB) subtypes of DLBCL, respectively [9]. A retrospective study from Italy on lenalidomide, an oral immunomodulatory agent, found an overall response rate (ORR) of 45.5% [10]. The combination of lenalidomide, ibrutinib, and the CD20 antibody, rituximab, in a phase Ib study, resulted in responses in 44% of patients with r/r DLBCL [11]. Everolimus, an oral novel agent inhibiting the raptor mammalian target of rapamycin (mTORC1) pathway, produced in a phase II study an ORR of 30% [12]. However, for all of these agents, the responses are not long-lasting, ranging from a few weeks to a couple of months. Therefore, more efficacious agents targeting new pathways are desperately needed.

## 3. CAR-T vs. ASCT

CAR-T is revolutionizing the management of patients with different types of lymphomas. CAR-T is a form of immunotherapy where the donor and recipient are the same. A patient with lymphoma denotes his own lymphocytes to be sent to a specialized manufacturing company, where the cells are genetically modified in vitro to express CAR. This modification enables the cells to track and eliminate CD19-positive lymphoma cells of the same patient in vivo. Different autologous CAR products have demonstrated long-lasting remissions in patient populations that are considered extremely difficult to manage and with a poor prognosis, such as those with r/r non-germinal center B cell (non-GCB), primary refractory patients, and those who are ineligible for intensive chemotherapy and ASCT. ZUMA-7 was a phase III trial which randomized patients with primary refractory or relapsed LBCL within one year to receive either the CAR-T product axicabtagene ciloleucel (axi-cel) or salvage chemotherapy followed by ASCT. The median event-free survival (EFS) after a median follow-up of two years showed the superiority of axicel (8.3 versus 2 months; HR 0.4) [13]. The TRANSOM trial was another interesting study with only minor differences in inclusion criteria in comparison to ZUMA-7. Lisocabtagene maraleucel (liso-cel) is a CAR-T product which, in the TRANSFORM trial, resulted in a better EFS in comparison with the ASCT arm (10.1 versus 2.3 months; HR 0.349) [14]. The BELINDA trial, which used tisagenlecleucel (tisa-cel) as another CAR-T product in a third randomized clinical trial, yielded negative results. The negative results of the BELINDA trial may be attributed to a combination of factors, such as the use of different CAR-T cell products (4-1BB CAR-T versus CD28), differences in population size, the proportion of patients with high-risk features, and the median time from leukapheresis to CAR-T cell infusion [15]. 

CAR-T therapy faces several challenges, including its complexity, lengthy manufacturing process, limited availability, and high cost, which make it difficult to standardize worldwide. In addition, the financial burden of CAR-T treatment can also pose a significant challenge for patients and healthcare systems. To address some of these issues, different allogeneic CAR-T products are currently being developed, with the goal of providing a more readily accessible off-the-shelf treatment option for patients with r/r disease [15,16].

ASCT has been the established standard of care for eligible patients with r/r DLBCL for many years and is considered a reliable option at centers with transplantation facilities. ASCT has been shown to produce significantly better five-year EFS (46% vs. 12%) and overall survival (OS) (53% vs. 32%) compared to conventional chemotherapy alone. However, ASCT has its limitations in patients with primary refractory and early-relapsing disease, the non-GCB subtype, and double- or triple-hit lymphoma [17,18]. ASCT is currently the best treatment approach for all eligible patients with r/r LBCL. However, the findings from the ZUMA-7 and TRANSFORM trials have highlighted the importance of categorizing patients into three distinct groups.

The first group includes patients with primary refractory disease or relapse within 12 months, who have a particularly poor outcome with standard treatment options, including ASCT. Recent data, generated from ZUMA-7 and TRANSFORM trials, show that CAR-T is superior to ASCT in managing this cohort [14,19].

The second group of patients with relapses beyond one year will continue to be subject to one of the established salvage chemotherapy regimens followed by high-dose chemotherapy and ASCT [20]. However, around 50% of patients undergoing ASCT will eventually relapse, and their prognosis is dismal [21,22].

The third group includes patients who are ineligible for ASCT. Reasons for ineligibility might be age, comorbidities, or refractoriness to salvage chemotherapy. For patients who fall into this group, the options for treatment may include assessing their suitability for CAR-T therapy, considering palliative chemotherapy, or exploring novel therapeutic agents.

Science is moving forward, and we are now exploring combining ASCT and CAR-T (ASCT-CAR-T) as a sequential treatment. Wang et al. retrospectively analyzed the data of patients with r/r DLBCL treated with ASCT (*n* = 46) versus a group of patients who received ASCT-CAR-T (*n* = 21). Despite reaching significantly higher CR (71% vs. 33%; *p* = 0.003) and progression-free survival (PFS) rates (80% vs. 44%; *p* = 0.036) with ASCT-CAR-T, the OS (80% vs. 69%; *p* = 0.545) remained similar [23], which indicates the need for other novel treatment strategies. 

## 4. Novel Antibodies

For several decades, rituximab has been the sole antibody approved for treating patients with CD20-positive lymphomas. Although rituximab leads to a significant improvement in the outcome of patients in first and subsequent lines of treatment, a considerable number of patients eventually become refractory to it, underscoring the need for new CD20-targeting antibodies. However, despite targeting different epitopes of the CD20 receptor than rituximab, newer antibodies, such as ofatumumab and obinutuzumab, have not shown any significant improvement over rituximab in DLBCL [24,25]. Multiple resistance mechanisms for anti-CD20-directed therapy have been proposed, and one of the primary mechanisms responsible for resistance in B-cell NHLs is the development of CD20-negative cells following treatment with anti-CD20-containing therapy. Scientists are continuously striving to improve the outcomes of patients with relapsed or refractory DLBCL by developing novel agents that target new molecules on the surfaces and inside the malignant B lymphocytes. Therefore, new surface antigens are being targeted (Figure 2), such as CD19, CD22, CD37, and CD79B. CD19 is considered a particularly promising target for the development of novel agents as it is highly expressed in all stages of B-cell maturation and in a majority of B-cell lymphoma types [26].

## 5. Tafasitamab

CD19 plays a crucial role in B-cell development, activation, and differentiation. Compared to CD20, the expression profile of CD19 is wider, and CD19 is expressed at an earlier pre-B stage, making it an attractive target in B-cell lymphomas [26]. CD19 shedding from the cell surface is not a common occurrence and is not typically observed, given that CD19 is a transmembrane protein that is typically anchored to the surface of B cells with regulated expression [27]. Therefore, antibodies targeting CD19 are less likely to be depleted or exhausted due to an absence of soluble CD19 in the circulation compared to antibodies targeting other B-cell surface markers. Although the native Fc CD19 antibodies displayed promising results in preclinical studies, they did not yield significant clinical success in the treatment of B-cell lymphomas. Clinical trials showed limited efficacy of this first generation of unmodified Fc CD19 antibodies. As a result, subsequent generations of CD19 antibodies have been engineered with modified Fc regions. Tafasitamab is a humanized monoclonal IgG1 antibody against CD19 that has been modified in its Fc region with two amino acid substitutions to enhance its affinity for Fcγ receptors. Antibody-dependent cellular cytotoxicity is mediated by the binding of tafasitamab’s Fc region to FcγRIIIa receptors on immune effector cells, which triggers the cells to release cytotoxic molecules such as perforin and granzyme B when combined with lenalidomide [28]. These molecules enter the lymphoma cells and activate signaling pathways that induce apoptosis, or programmed cell death. This process is a key mechanism of action of the combination of tafasitamab and lenalidomide, and this interaction between the two medications is thought to play a major role in their antitumor activity.

Tafasitamab was studied in a single-arm, phase II study (L-MIND) in patients with r/r DLBCL who are ineligible for ASCT. According to the results of the L-MIND study, 60% of patients (*n* = 80) with r/r DLBCL who were treated with a combination of tafasitamab and lenalidomide achieved an objective response, with 43% achieving CR, and the median duration of response, 22 months, was very encouraging [29]. After ≥35 months, the median duration of response, median PFS, and median OS were 43.9, 11.6, and 33.5 months, respectively. More than 80% of patients who achieved CR remained in remission after 3.5 years [30]. Recent matching-adjusted indirect comparisons of the results of the L-MIND study with polatuzumab-BR regimen found that tafasitamab–lenalidomide was associated with a significantly longer duration of response (DOR) (HR 0.34 [95% CI 0.12, 0.98]; *p* = 0.045), improved OS (HR 0.39 [95% CI 0.18, 0.82]; *p* = 0.014), and PFS (HR 0.39 [95% CI 0.29, 0.53]; *p* < 0.001) [31]. Real-world data from Italy with 76 patients found an ORR of 67.1% for tafasitamab–lenalidomide [32].

Rituximab and lenalidomide have demonstrated synergistic effects by enhancing the immune response and directing activated T cells to the tumor microenvironment. However, it is uncertain if the addition of rituximab to tafasitamab and lenalidomide would further improve treatment efficacy. Additionally, the decision to add rituximab should be weighed against potential risks of greater toxicity. An important point to consider is that tafasitamab is presently approved for use with lenalidomide, and we know that lenalidomide discontinuation rates in patients with DLBCL can be substantial. 

Among patients enrolled in the L-MIND study, neutropenia (48%), thrombocytopenia (17%), and febrile neutropenia (12%) were the most common grade ≥ 3 AEs. Serious AEs were observed in 51% of patients, with pneumonia (6%), febrile neutropenia (6%), pulmonary embolism (4%), bronchitis (2%), atrial fibrillation (2%), and congestive cardiac failure (2%) being the most frequently reported serious AEs [29].

Tafasitamab is currently being investigated in various clinical trials for its potential as a first-line treatment option for DLBCL. The FrontMIND study, a phase III randomized trial, is evaluating tafasitamab in combination with lenalidomide and rituximab, cyclophosphamide, doxorubicin, vincristine, and prednisone (R-CHOP), as compared to R-CHOP alone, in untreated high-intermediate-risk and high-risk patients with DLBCL (NCT04824092). Additionally, an upfront trial is exploring the use of tafasitamab–lenalidomide in combination with rituximab in elderly patients aged 80 years or older (NCT04974216). Another phase II study, known as the Smart Stop Study, is investigating a novel regimen involving rituximab, lenalidomide, acalabrutinib, and tafasitamab, with and without chemotherapy, in previously untreated patients with non-GCB DLBCL (NCT04978584). In the relapsed setting, a promising chemotherapy-free combination of mosunetuzumab, polatuzumab, tafasitamab, and lenalidomide is currently being evaluated (NCT05615636). The ongoing B-MIND study (NCT02763319) is a phase II/III randomized trial that is evaluating tafasitamab and rituximab plus bendamustine in patients with r/r DLBCL who are not eligible for ASCT. Lastly, the safety and effectiveness of tafasitamab and lenalidomide, in combination with plamotamab, an investigational bispecific antibody that targets CD20 and CD3, is being studied in patients with r/r DLBCL (NCT05328102).

## 6. Checkpoint Inhibitors

Checkpoint inhibitors (ChIs) showed high efficacy in classic Hodgkin lymphoma. However, studies with single-agent nivolumab or pembrolizumab in combination with RCHOP or post-ASCT failed to show significant outcome improvement in aggressive lymphomas [33,34,35]. Nonetheless, pembrolizumab seems to be effective in patients with primary mediastinal B-cell lymphoma (PMLBCL). In a phase Ib trial, KEYNOTE-013, 21 patients had an objective response rate of 48% (CR: 33%), while in a phase II trial, KEYNOTE-170, 53 included patients had an objective response rate of 45% (CR: 13%). Median PFS and OS were 5.5 and 22.3 months, respectively [36]. During the 2021 ASH annual meeting, the final analysis of KEYNOTE-170 showed a prolonged DOR, with a median DOR that was not reached after a median follow-up of 48.7 months [37].

Pembrolizumab is well tolerated, as observed in previous trials with grade ≥ 3 AEs occurring in 22.6% of patients, while neutropenia was observed in only three patients [37]. 

Pembrolizumab and nivolumab are currently being investigated in both upfront and relapsed settings for the treatment of DLBCL. One phase II study (NCT03995147) is exploring the use of pembrolizumab in combination with R-CHOP as a first-line treatment for patients with high-risk, untreated non-GCB DLBCL. Nivolumab plus chemoimmunotherapy versus chemoimmunotherapy alone is also currently being studied in a phase III trial in patients with newly diagnosed PMLBCL (NCT04759586). In the r/r setting, we have an ongoing phase II randomized study (NCT05221645) comparing the efficacy of R-ICE versus pembrolizumab-R-ICE in patients with r/r DLBCL. In addition, one study (NCT02446457) is investigating the potential benefit of adding pembrolizumab to rituximab with or without lenalidomide. Lastly, a phase II clinical trial is underway to evaluate the efficacy of brentuximab vedotin and nivolumab, either alone or in combination with R-CHOP, for the treatment of patients with PMBCL who have not yet received treatment (NCT04745949).

## 7. Magrolimab

CD47 is highly expressed on numerous types of cancer cells, including lymphoma cells, where it serves as an antiphagocytic molecule. CD47 enables cancer cells to avoid being phagocytosed and destroyed by macrophages, thereby facilitating their evasion of the immune system. The overexpression of CD47 has been identified as an independent predictor of an unfavorable prognosis in patients with different types of cancer, such as lymphoma [38]. CD47 is emerging as a promising and potent macrophage immune checkpoint. Magrolimab is a humanized anti-CD47 IgG4 antibody that inhibits the CD47 signaling pathway. It induces phagocytosis of tumor cells via the blockade of CD47 and its ligand SIRPα [39]. Magrolimab was tested in a phase Ib study with rituximab in 15 patients with r/r DLBCL. A significant number of patients responded, with 40% achieving an objective response and 33% achieving CR. After six months of treatment with magrolimab, responses were maintained, with over 90% of patients experiencing sustained improvement [40]. The interim analysis of the ongoing phase Ib trial (NCT02953509) presented at the ASH2022 annual meeting showed that, over a median follow-up of 11.3 months (range, 0.1–33.4), patients with r/r DLBCL (*n* = 33) achieved an ORR of 51.5%, which included 39.4% with CR, when treated with the combination of magrolimab and rituximab plus gemcitabine and oxaliplatin (R- GemOx). Of note, the median DOR among responders was 18 months. The 1-year PFS rate was 35.4% (95% CI, 17.5–53.9%), and the 1-year OS rate was 67.1% (95% CI, 47.5–80.8%). These results suggest that combination therapy may be a promising treatment option for patients with r/r DLBCL [41].

Nearly all patients (93.9%) experienced grade ≥ 3 AEs. The most frequent AEs observed were anemia (≥grade 3: 60.6%), thrombocytopenia (≥grade 3: 42.4%), and neutropenia (≥grade 3: 18.2%) [41].

Currently, there are no active clinical trials registered under clinicaltrials.gov investigating magrolimab in LBCL, suggesting a lack of current interest in further evaluating this agent in this context, which might be attributed to its toxicity profile.

## 8. Antibody–Drug Conjugates

Antibody–drug conjugates (ADCs) are a group of monoclonal antibodies directed against a specific cell surface structure which is usually overexpressed on malignant cells. They are designed to be a carrier to deliver the cytotoxic agent to the target cells with the least off-target effects. ADCs are composed of antibodies conjugated through a chemical linker with a toxic drug. The linker must maintain the balance between stability and lability to ensure that the cytotoxic payload is released only inside targeted malignant cells, while minimizing damage to healthy tissue. Once the antibody binds to the targeted receptor on the surface of malignant cells, it induces endocytosis of the receptor with the adherent antibody, including the poison. With this approach, we are able to smuggle higher concentration of chemotherapy inside the malignant cells and minimize systemic side effects. Examples of receptors which are currently being targeted in hematological malignancies are CD19, CD22, CD33, CD30, BCMA, and CD79b [42].

## 9. Polatuzumab Vedotin

Polatuzumab vedotin (polatuzumab) is a so-called third-generation ADC. It is a fully humanized IgG1 antibody targeting CD79b which is part of the B-cell signaling pathway and expressed exclusively on B lymphocytes. It is overexpressed on different B-cell lymphomas. Polatuzumab is linked to the microtubule polymerization inhibitor, the highly toxic monomethyl auristatin E (MMAE). Polatuzumab was tested in combination with rituximab in a phase II trial (ROMULUS) in difficult-to-treat patients with r/r DLBCL with four median prior lines, showing an ORR and a CR rate of 54% and 21%, respectively [43]. The first approval of polatuzumab was based on a phase Ib/II trial (GO29365 study) in combination with rituximab and bendamustine (polatuzumab-BR) in patients with r/r DLBCL. Patients in the GO29365 study received in median two prior lines, with 46% of patients failing three lines and 53% of patients being primary refractory. In the randomized part of the study (*n* = 80), the polatuzumab-BR arm reached, in comparison with the BR arm, significantly higher CR rate (40.0% vs. 17.5%; *p* = 0.026), longer PFS (median, 9.5 vs. 3.7 months; HR, 0.36, 95% CI, 0.21 to 0.63; *p* < 0.001), and longer OS (median, 12.4 vs. 4.7 months; HR, 0.42; 95% CI, 0.24 to 0.75; *p* = 0.002; median follow-up, 22.3 months) [44]. After an additional 27 months, the best response rate was 52.5%. In the extension cohort with 108 patients treated with polatuzumab-BR, the ORR was 52.8% [45]. Hematological toxicity was more pronounced in the polatuzumab-BR arm. However, the rate of grade 3–4 infection complications was similar. The most common grade 3 and 4 adverse events (AEs) in the polatuzumab arm were neutropenia (40%), anemia (11%), and peripheral neuropathy (9%).

The real-world data are also encouraging. In a retrospective German study with 105 patients with r/r LBCL treated with a polatuzumab-containing regimen, 41 patients received polatuzumab as a bridge to CAR-T. Bridging was successful in 51.2% of patients, attaining a 6-month OS of 77.9% in the entire cohort treated with polatuzumab. Combining polatuzumab and rituximab without bendamustine resulted in an ORR of 40%. Polatuzumab was also effective post-CAR-T, with 58.3% of patients that failed CAR-T responding to a polatuzumab-containing regimen [46]. In anotherretrospective analysis conducted in the United States with 57 patients refractory to CAR-T, polatuzumab–rituximab (*n* = 54) and polatuzumab-BR (*n* = 35) showed cumulative CR in 14% with an ORR of 44% [47]. Different real-world data collections from different countries showed an ORR ranging from 47.9 to 71.4% and a CR rate of 18.2–40% [48,49,50,51]. The largest volume of real-world data comes from the United Kingdom, with 133 patients treated with polatuzumab-BR resulting in an ORR of 57.0% (CR, 31.6%) [52].

Polatuzumab has also reached the upfront setting. In the POLARIX phase III trial, polatuzumab was tested in combination with rituximab, cyclophosphamide, doxorubicin, and prednisone (polatuzumab-RCHP) versus R-CHOP. The polatuzumab-RCHP arm had a significantly lowered risk for progression, relapse, and death (stratified hazard ratio, 0.73; 95% confidence interval [CI], 0.57 to 0.95; *p* = 0.02) [53]. However, the magnitude of benefit in PFS was limited, with only a 6.5% improvement compared to standard-of-care R-CHOP, and there was no OS improvement in the potential curative setting. This benefit must be balanced against the significant increase in cost in the upfront setting. In particular, the lack of benefit of polatuzumab-RCHP in different subgroups of patients with DLBCL, such as those who are ≤60 years of age, have the GCB subtype, double- or triple-hit, bulky disease, and an IPI score ≤ 2, may limit its application as the new standard of care in frontline treatment.

The current approved partner regimen for polatuzumab in patients with r/r lymphoma is BR. However, bendamustine, a component of BR, has a long wash-out time of at least 12 weeks, which may be problematic for patients who are scheduled for CAR-T therapy. Many r/r lymphoma patients cannot be left without treatment for such a long period, and this may prompt physicians to use other non-bendamustine/polatuzumab-containing regimens prior to leukapheresis for CAR-T. Therefore, the current approval of polatuzumab with only BR may limit its widespread use in r/r lymphoma patients who are candidates for CAR-T.

At present, polatuzumab is being studied in a phase III study in combination with rituximab, gemcitabine, and oxaliplatin (polatuzumab-R-GemOx) compared to R-GemOx in participants with r/r DLBCL (POLARGO; #NCT04182204) and in another phase II trial with a similar population of patients in combination with rituximab, ifosfamide, carboplatin, and etoposide (polatuzumab-R-ICE) versus R-ICE (#NCT04665765). A phase II study is evaluating the safety and efficacy of polatuzumab in combination with R-CHP as an upfront treatment in patients with double- and triple-hit lymphoma, double-expressor lymphoma, and high-grade B-cell lymphoma (NCT04479267). Furthermore, polatuzumab is being studied as a consolidation treatment in ASCT recipients that have B-cell NHL and have reached CR, PR, or stable disease after ASCT (NCT04491370).

## 10. Pinatuzumab Vedotin

Pinatuzumab vedotin (pinatuzumab) is a humanized IgG1 ADC which is directed against CD22 and linked with MMAE via a protease-cleavable peptide. CD22 is expressed on mature B-cell lymphocytes and plays a role in B-cell receptor signaling [54]. Pinatuzumab has demonstrated encouraging clinical activity as a single agent and in combination with rituximab. Patients with r/r DLBCL treated with 2.4 mg/kg pinatuzumab in a phase I trial had an ORR of 36% (9/25). The CR rate of patients treated with single-agent pinatuzumab and pinatuzumab in combination with rituximab (R-pinatuzumab) was 16% and 25%, respectively [54]. In another phase I/II study, R-pinatuzumab induced in the r/r DLBCL cohort a 60% ORR and 26% CR rate. Median OS, PFS, and DOR were 16.5, 5.4, and 6.2 months, respectively. Grade 3–5 AEs were frequent (78.6%), leading to treatment discontinuation in 42.9% of patients and dose reduction in 14.3% [43]. The abovementioned ROMULUS study showed the greater efficacy and better tolerability of polatuzumab over pinatuzumab, and therefore, the manufacturer opted for further development of polatuzumab and ceased conducting tests with pinatuzumab.

## 11. Naratuximab Emtansine

Naratuximab emtansine (naratuximab) is a humanized IgG1 ADC recognizing CD37 that is conjugated via a thioether-based linker to a cytotoxic maytansinoid, DM1. CD37 is a surface antigen overexpressed on B-cell lymphocytes. Its function has not been fully elucidated. However, it plays a significant role in immune regulation and tumor suppression. In a phase II trial with 80 patients with r/r DLBCL, naratuximab and rituximab produced an ORR of 43.2% and a CR rate of 32.4%. However, 82% experienced grade ≥ 3 AEs, with neutropenia (54%), anemia (17%), and thrombocytopenia (11%) being the most-documented higher-grade AEs [55].

As of the time of writing this manuscript, there are no ongoing clinical trials of naratuximab for the treatment of lymphoma registered on clinicaltrials.gov. It is important to note that this does not necessarily mean that there are no ongoing clinical trials of naratuximab for other indications or in other countries that may not be included on this database.

## 12. Zilovertamab Vedotin

Zilovertamab vedotin (zilovertamab) is a humanized ADC that is conjugated to MMAE and targets extracellular receptor tyrosine kinase-like orphan receptor 1 (ROR1). ROR1 is a cell surface protein on fetal cells which is usually not expressed on normal cells after birth. However, some solid and hematological cancer cells regain the ability to express ROR1, which promotes the migration and survival of malignant cells. A phase I study with 13 patients with r/r DLBCL proved its efficacy as a single agent, obtaining an ORR of 38.5% and a CR rate of 23.1%. Treatment-related AEs were observed in 47.1% of the cohort. The most commonly observed ≥3 AEs were neutropenia (29.4%), anemia (15.7%), febrile neutropenia (7.8%), peripheral neuropathy (7.8%), platelet count decrease (7.8%), diarrhea (5.9%), and pneumonia (5.9%) [56].

Zilovertamab is currently being tested in a phase II/III study in combination with R-CHP for participants with untreated DLBCL (NCT05139017) and as a single agent in two phase II studies on patients with r/r DLBCL (NCT05406401 and NCT05144841). 

## 13. Loncastuximab Tesirine

Loncastuximab tesirine (loncastuximab) is an ADC composed of a humanized monoclonal IgG1 antibody, a linker, and a potent cytotoxic agent. The antibody is directed against the CD19 receptor and conjugated to a pyrrolobenzodiazepine dimer cytotoxin (PBD), SG3199 [57]. The short half-life of this payload reduces the risk of excessive systemic toxicity [58]. CD19 has emerged as an attractive target in several B-cell malignancies as it is highly present on the surface of B-cell lymphocytes throughout all stages of their maturation [59]. 

In a multi-cohort phase I study (LOTIS-1), 139 patients with r/r DLBCL were treated with loncastuximab. The ORR of the DLBCL cohort was 42.3%, with 23.4% reaching CR. However, the PFS was just 3.1 months [60]. Moreover, a recent subsequent phase II study evaluated the efficacy and safety of loncastuximab in patients with r/r DLBCL who failed two or more prior lines. The study included 145 patients and reported an ORR of 48.3%, with an encouraging CR rate of 24.1%. The most common grade ≥ 3 treatment-related AEs were neutropenia (26%), thrombocytopenia (18%), and increased gamma-glutamyltransferase (17%); 39% of patients experienced serious AEs [57].

Further research is needed to better understand the long-term safety and efficacy of loncastuximab, particularly in combination with other agents.

Loncastuximab is currently under investigation in patients with LBCL in various settings and populations. In the upfront setting, it is being studied in combination with DA-EPOCH-R for individuals with aggressive disease, such as double-hit DLBCL (NCT05270057) and as a single agent in combination with rituximab for untreated frail patients (NCT05144009). In the r/r setting, it is being investigated in combination with rituximab compared to chemoimmunotherapy (R-GemOx) (NCT04384484) and in combination with chemotherapy before ASCT (NCT05228249). Moreover, loncastuximab is being examined as a maintenance therapy in high-risk patients following ASCT (NCT05222438) and CAR-T (NCT05464719).

## 14. Bispecific Antibodies (BsAbs) and Bispecific T-Cell Engagers (BiTE)

BsAbs are immunoglobulins that are directed against two different antigens. They are designed to bring CD3-positive T cells nearby to malignant cells through binding to a tumor-specific antigen, resulting in T-cell activation and death of the cancer cells. BsAbs are composed of a combination of various antibody fragments joined together by flexible linker. BsAbs can vary in their structure and may include full-sized IgG and other, more intricate formats. Bispecific T-cell engagers (BiTEs) are a subclass of BsAbs that are composed of two single-chain variable fragment (scFv) regions, which are derived from monoclonal antibodies [61]. The two scFvs are joined by a flexible peptide linker. BiTEs are not fully antibodies. They are smaller and simpler than other BsAbs. 

Unlike regular antibodies, BsAbs seem to activate T cells without the need for a co-stimulatory signal. BsAbs hold a significant advantage over CAR-T as they are easily accessible as off-the-shelf products, unlike autologous CAR-T, which requires several weeks to be ready for re-infusion. Furthermore, BsAbs have lower toxicity in comparison with other treatment approaches (chemotherapy, transplantation, CAR-T). However, it is not clear if BsAbs will exhibit the same long-lasting remissions observed with other comparable treatment approaches. A more comprehensive understanding of the efficacy, potential long-term side effects, and where the treatment fits in the sequence of management actions for patients with LBCL can be attained through a new generation of clinical studies, extended follow-up, and broader real-world experience.

## 15. Glofitamab

Glofitamab is a fully humanized IgG1-bispecific antibody that has a unique binding pattern of two CD20 binding domains and one CD3 binding domain. By binding to both CD20 on B cells and CD3 on T cells, it brings the two cell types closer together, triggering the activation of T cells. This activation leads to a release of cytotoxic molecules and cytokines, which further enhance T-cell migration and proliferation, ultimately promoting the elimination of malignant CD20-expressing B cells [62]. The intensity of CD20 expression does not appear to be a reliable predictor of response to glofitamab [62].

Glofitamab was studied in an interesting study in an upfront setting. Patients with newly diagnosed DLBCL were treated in a phase I study with R-CHOP (first cycle) followed by glofitamab-R-CHOP in subsequent cycles (6–8 cycles in total). After a median follow-up of 5.6 months (range: 5.1–10.3), the ORR was 93.5% (43/46), with 76.1% (35/46) of the participants achieving CR [63]. A phase I trial studied the combination of obinutuzumab and glofitamab in 127 patients with aggressive B-cell lymphoma. The ORR and CR rate were 48% and 33.1%, respectively [64]. In a phase II study with 154 patients with r/r LBCL, at a median FU of 12.6 months, glofitamab was associated with an ORR and CR rate of 52% and 39%, respectively. In the population of patients who failed CAR-T (*n* = 52), glofitamab induced CR in 35%. The median time to CR was exceptional (42 days) in a difficult-to-treat cohort. The 12-month PFS was 37% (95% CI, 28 to 46). Altogether, 78% of patients maintained their CR status after 12 months, which is very encouraging [65]. According to the latest published data, 46% and 20% of patients completed the 18-month and 42-month post-treatment FUs, respectively. The majority of patients who achieved CR at the end of treatment (45/61; 74%) remained in CR [66].

Cytokine release syndrome (CRS) was the most commonly reported AE, with a prevalence of 63%, although only 3% of patients experienced grade ≥ 3 CRS. Grade ≥ 3 AEs were observed in 62% of patients, including grade ≥ 3 neurologic events in 3% [65]. When combined with R-CHOP, grade ≥ 3 AEs occurred in 71.4% of patients, with 23.2% of grade ≥ 3 AEs attributed to glofitamab. Serious AEs associated with glofitamab were found in 8.9% of patients. Three patients experienced grade 5 AEs, two of which were due to COVID-19 pneumonia [63].

Glofitamab is currently under investigation for use in previously untreated patients with aggressive B-cell lymphoma. Ongoing studies, such as COALITION, are examining the effectiveness of combining glofitamab with R-CHOP (NCT04980222) or polatuzumab-RCHP chemotherapy regimens for younger patients with higher-risk DLBCL or high-grade B-cell lymphoma (NCT04914741) and rituximab plus polatuzumab (NCT05798156) in R-CHOP-ineligible patients. In r/r lymphoma, glofitamab is being studied in combination with the JAK2 inhibitor poseltinib and lenalidomide (NCT05335018), as well as with investigational cereblon inhibitors CC-220 and CC-99282 (NCT05169515). Several additional studies are ongoing in the r/r setting, including investigating glofitamab in combination with R-ICE (NCT05364424) or GemOx (NCT04313608). 

## 16. Mosunetuzumab

Mosunetuzumab is a bispecific full-length humanized IgG1 antibody that targets CD20 and CD3. The variable domains of the heavy and light chains are responsible for binding to CD3 and CD20, respectively. By targeting both proteins, mosunetuzumab can recruit T cells to the B-cell lymphoma tissue facilitating T-cell-mediated killing of the lymphoma cell. The altered Fc region lacking glycans leads to the loss of its effector function [67,68]. A phase I/II study conducted on 129 patients diagnosed with relapsed/refractory aggressive B-NHL revealed an ORR of 34.9%. The median DOR and PFS were 7.6 months and 1.4 months, respectively. Similar data were observed in patients who relapsed after CAR-T. Although the CR rate was limited to 19.4%, the duration of CR was found to be relatively long-lasting, with an estimated median of 22.8 months. Even 16 months after discontinuation of mosunetuzumab, 70.8% of patients remained progression-free [69]. In an interesting phase I/II approach to managing elderly patients with DLBCL who were not fit for standard chemotherapy, 54 patients received mosunetuzumab as an upfront regimen. The ORR and CR rate were 56% and 42%, respectively. The one-year PFS rate was 39% (95% CI: 25.8–52.8); there were no reported cases of immune-effector-cell-associated neurotoxicity syndrome (ICANS) related to mosunetuzumab [70]. Preliminary data from a phase Ib/II clinical trial, which investigated the addition of mosunetuzumab to R-CHOP in 36 previously untreated patients with DLBCL, showed an ORR of 96% and a CR rate of 85%. However, the trial also documented grade ≥ 3 AEs in 86% of patients, and serious AEs in 44% [71].

As a single agent, mosunetuzumab led to grade ≥ 3 AEs in 71.1% of patients, with neutropenia (25.4%), hypophosphatemia (15.2%), and anemia (9.1%) being the most frequent (≥5%) [69]. CRS occurred in 27.4%; out of these grade 3 was observed in just 1% but no grade 4 were reported [69].

Ongoing clinical trials are evaluating the safety and efficacy of mosunetuzumab in the treatment of r/r aggressive lymphoma. Mosunetuzumab is being tested in combination with GemOx (NCT04313608), polatuzumab (NCT03671018 and NCT05171647), a platinum-based salvage regimen prior to ASCT (NCT05464329), and as a consolidation therapy post-ASCT (NCT05412290).

## 17. Epcoritamab

Epcoritamab is a subcutaneously IgG1-bispecific antibody redirecting CD3-positive T lymphocytes to CD20-positive B-cell lymphoma tissue. In the EPCORE NHL-1 phase II trial with 157 patients with r/r LBCL, the ORR of patients treated with single-agent epcoritamab was 63.1%, with 38.9% achieving CR. The median DOR was 12 months, with a DOR of 9.7 months in the post-CAR-T cohort. Additionally, 45.8% achieved MRD negativity, as determined using a ctDNA NGS assay [72]. According to the updated results of a phase I/II study with 29 patients treated with epcoritamab in combination with rituximab, dexamethasone, cytarabine, and oxaliplatin or carboplatin (R-DHAX/C), the ORR and CR rate were 100% and 80%, respectively. Eleven patients decided to continue with epcoritamab rather than undergo ASCT, and 45% of them reached complete metabolic remission. Their median DOR was not reached [73].

Grade 3–4 neutropenia and grade 3 anemia were found in 14.6% and 10.2% of patients, respectively. CRS occurred in 49.7% with only 2.5% having grade 3. Ten patients experienced ICANS events, with nine of the events being grade 1 or 2 and one event being grade 5 [72].

Multiple clinical studies are currently underway to assess the safety and efficacy of epcoritamab in patients with aggressive lymphoma, both in the first and subsequent lines of treatment. For instance, EPCORE DLBCL-1 and EPCORE DLBCL-2 are phase III trials that are evaluating the use of epcoritamab in combination with R-CHOP as a first-line treatment (NCT04628494 and NCT05578976). Another phase II study is exploring the use of epcoritamab in combination with lenalidomide as a first-line treatment in elderly patients ineligible for anthracycline (NCT05660967). Additionally, other trials are ongoing to evaluate the effectiveness of epcoritamab as a single agent in r/r LBCL (NCT04358458 and NCT05451810).

## 18. Odronextamab

Odronextamab is a fully human IgG4-bispecific antibody which targets both CD20 on B cells and CD3 on T cells, leading to the activation and engagement of T cells against malignant B cells. A single-arm phase I study (ELM-1), which treated 45 patients with r/r DLBCL with odronextamab, demonstrated an ORR and CR rate of 40% and 35.6% in all population with 33% and 27% in patients who had previously received CAR-T, respectively [74]. The phase II ELM-2 study enrolled 121 patients with r/r DLBCL. After a median FU of 17.1 months, the ORR and CR rate were 53% and 37%, respectively. The median DOR was not reached [75]. In a dose expansion trial in 10 patients who received ≥80 mg odronextamab, the ORR and CR rate were 60% with a median PFS of 11.1 months [76].

With the step-up dosing, grades 1 and 2 CRS were observed in 46% of patients; no grade ≥ 3 AEs were seen; and ICANS occurred in 4% of patients [75].

At present, two clinical trials are ongoing to further investigate the role of odronextamab in r/r DLBCL (phase II: NCT03888105 and phase I: NCT02290951).

## 19. Blinatumomab

Blinatumomab is a first-in-class BiTE which consists of anti-CD19 and anti-CD3 domains allowing T cells to recognize CD19-positive lymphoma cells and destroy them. Blinatumomab is not a full-length antibody and lacks, like other BiTEs, the Fc region. Therefore, the half-life of blinatumomab is as short as 2–4 h. This is in contrast to the longer half-life observed with full-length monoclonal antibodies, which is attributed to the recycling process mediated by Fc receptors [77]. Blinatumomab demonstrated efficacy in r/r DLBCL in two separate phase II trials with 21 and 41 patients, with a response rate of 43% and 37%, including CR rates of 19% and 22%, respectively [78,79]. In a pooled analysis of three phase I/II studies that included the aforementioned two phase II trials and a total of 73 patients with r/r DLBCL, 23% of patients achieved CR [80]. There are other data showing moderate efficacy of blinatumomab when combined with lenalidomide or R-CHOP or when given as a consolidation treatment after first-line ASCT or allogeneic stem cell transplantation [80,81,82,83].

Grade ≥ 3 AEs were observed in 71% of patients, with grade 3 neurologic AEs found in 24% of the treated patients [79].

Currently, there are limitations to conducting clinical trials with blinatumomab as a treatment for DLBCL due to several factors. One of the main challenges is blinatumomab’s short half-life, which necessitates continuous infusion via a central line for several days. This mode of administration can pose practical challenges and may limit the widespread use of blinatumomab in clinical settings. Moreover, it is important to note that the efficacy of blinatumomab’s short half-life in reaching soluble and rapidly growing acute lymphoblastic leukemia cells may not be generalizable to all lymphomas. Further research is needed to investigate the optimal dosing and delivery strategies of blinatumomab for different types of lymphomas, including DLBCL.

## 20. Others

Besides the novel antibodies mentioned in Table 1, there have been various other novel antibodies that have not shown sufficient efficacy or have demonstrated excessive toxicity, whereas others are still in the earlier stages of research, such as plamotamab (BsAbs CD20-CD3), which resulted in an ORR of 41% and a CR rate of 26% in 27 patients with DLBCL [84]; ADCs, such as camidanlumab tesirine (anti-CD25; CD25 is part of the IL-2 receptor), which resulted in an ORR of 31% and a CR rate of 12.5% in 16 patients with LBCL [85]; and inotuzumab ozogamicin, which resulted in an ORR of 51.4% in 72 patients with LBCL (42 DLBCLs and 30 other aggressive lymphomas) [86], but the results were disappointing in a phase III randomized trial [87].

## 21. Resistance Mechanisms, Challenges, and Future Perspectives

To date, our understanding of the mechanisms underlying resistance to novel antibodies is limited due to the paucity of available research in this area. However, recent data, primarily consisting of those from preclinical studies, suggest that LBCL cells have the ability to evade the effects of novel antibodies including ADCs, BsAbs, and BiTEs through various mechanisms (Table 2). 

For instance, in vitro, downregulation, loss of expression, or mutation in CD79b may reduce the efficacy of polatuzumab [88]. However, other in vitro studies in animals and humans found no strong correlation between a high density of expression of CD79b or CD22 and the cytotoxic activity of ADCs, suggesting that there may be other factors than target overexpression that play a role in conferring sensitivity to ADCs [89,90]. Another way of escaping ADCs is the development of efflux pumps that can expel various toxic compounds. The overexpression of the multidrug resistance pump (MDR1) can reduce the effectiveness of payloads such as MMAE, resulting in resistance of lymphoma cells. This has led to investigations into several drug efflux pump inhibitors to address this issue [91]. Furthermore, activation of compensatory signaling pathways that bypass the targeted pathway reduces the effectiveness of ADCS. Some DLBCL cells may downregulate immune checkpoint proteins or upregulate inhibitory proteins, making them resistant to immune-mediated cytotoxicity induced by the novel antibody [42]. Resistance can also arise from the tumor microenvironment through processes such as hypoxia and vascularization, which can restrict the penetration and distribution of ADCs in lymphoma tissue.

Strategies to overcome resistance include the development of novel ADCs with different linkers or payloads, the combination of ADCs with other therapies, such as checkpoint inhibitors, and the use of biomarkers to identify patients who are more likely to respond to ADCs [92,93]. Additionally, in vitro studies have shown that the combination of ADCs with agents that target the tumor microenvironment or signaling pathways can overcome resistance and improve responses [94,95]. Moreover, the application of novel payloads such as inhibitors of nicotinamide phosphoribosyltransferase (NAMPT) in the treatment of T-cell lymphomas could potentially decrease the occurrence of adverse events, including neurological side effects that may arise from the use of microtubule inhibitors [96]. Understanding the mechanisms of resistance to ADCs in lymphoma is crucial for the development of effective therapies and more advanced strategies.

CD20 is known to be shed from the surface of B cells into the circulation, leading to the formation of soluble CD20, which can bind and deplete CD20 antibodies, potentially reducing their efficacy over time [97]. One way to potentially enhance the efficacy of CD20 antibodies is to administer an anti-CD20-directed therapy prior to the infusion of CD20-based BsAbs. Another approach would be to use modified CD20 antibodies that have a higher binding affinity for CD20 and are less affected by the presence of soluble CD20.

Loss of target antigens and receptor gene mutation after previous CD20-directed therapy may contribute to resistance against anti-CD20-directed therapy [98]. Additionally, mutations in TP53 [99,100] and MYC amplification have been detected in multidrug-resistant patients, and these alterations are recognized to contribute to the resistance to biologic therapy [101,102]. The murine double minute-2 (MDM2) inhibitor may help in this context, and has just entered clinical studies [103].

Furthermore, studies have indicated that T lymphocytes in the tumor microenvironment may exhibit downregulation of CD3 on their surface, which can lead to resistance to anti-CD3-directed therapy. This phenomenon is known as T-cell exhaustion, which occurs as a result of persistent T-cell activation, leading to the upregulation of inhibitory markers and a reduced ability of T cells to secrete proinflammatory cytokines, which are crucial for an effective antitumor response [104]. 

Recent advancements in the Azymetric and THIOMAB platform technologies have led to the development of several promising therapeutic antibodies. These technologies optimize the design of monoclonal antibodies to improve their half-life, affinity, and efficacy, resulting in antibodies with improved pharmacological properties [105,106].

Another significant challenge which is not attributed to resistance is posed by AEs, for instance, CRS and ICANS, which are usually more common in anti-CD19-based novel antibodies. Recent research identified brain mural cells expressing CD19 that might be attacked by CD19-CAR-T [107]. This can lead to increased vessel permeability causing inflammation and a release of proinflammatory molecules such as VLA4, interleukin (IL) 1 and 6, and GM-CSF, which may be responsible for the CNS symptoms of patients. On the other hand, CD22 was also found on brain glia cells [108]. However, anti-CD22-based BsAbs do not seem to be associated with a higher ICANS occurrence. To limit the complications of CRS, we have been using tocilizumab as an IL-6 inhibitor, which is associated with an increase in the occurrence of ICANS. As an alternative, current research is exploring siltuximab, an inhibitor of the soluble IL-6 receptor, which limits the increase in soluble IL-6 in CSF observed during treatment with tocilizumab and plays a role in the development of ICANS [109]. The anti-GM-CSF lenzilumab, the VAL4 antibody natalizumab, and several JAK inhibitors might help in reducing the occurrence of ICANS [110,111,112]. Lastly, IgG1 and 3 antibodies used in different novel antibodies can trigger the complement cascade in opposition to IgG2 and 4. Therefore, using IgG4 and also IgM might reduce the CRS rate [95].

## 22. Factors That May Affect the Decision

Due to the availability of several good options for the treatment of r/r patients with LBCL, different factors may affect the decision. One of the most critical questions to address in the future would be whether CAR-T therapy should be prioritized over novel antibody-containing regimens. While there are promising long-term data emerging for CAR-T therapy, there are also encouraging results from novel antibody therapies such as tafasitamab (used in combination with lenalidomide), which have demonstrated a plateau in the PFS curve for up to 3–4 years [30]. However, CAR-T therapy is associated with a considerable side effect profile, such as CRS and ICANS, and the eligibility criteria can be restrictive. On the other hand, novel antibodies are better tolerated than CAR-T but can have limitations based on the antigen expression profile of the tumor. Given the significant potential side effects associated with CAR-T therapy, CAR-T may be less suitable for elderly patients or individuals with pre-existing comorbidities, which represent a considerable fraction of patients with LBCL. In such cases, novel antibodies may provide a viable alternative treatment approach. Additionally, it is necessary to consider and prepare for future treatment options. If CAR-T therapy is being considered, it may be advisable to avoid antibodies targeting CD19, as they may be less effective after CAR-T. Conversely, anti-CD19-containing regimens may be less effective after CAR-T because of the reported loss of CD19 and resistance to antibodies targeting CD19 signaling after CAR-T [113,114]. Additionally, the majority of patients undergoing CAR-T will relapse, and there is no standard of care for managing these patients. Nonetheless, there are some limited data showing the encouraging efficacy of CD-19 antibodies post-CAR-T [60,69]. In addition, CD19 expression was still present after tafasitamab-containing treatment [29], and some patients who received loncastuximab responded to subsequent treatment with CAR-T [60].

As previously discussed, the period since the last treatment is also crucial because of the required wash-out time of some agents prior to T-cell apheresis for CAR-T manufacturing. The duration of the required wash-out period prior to T-cell apheresis for CAR-T manufacturing can vary depending on the specific chemotherapy protocol, with some protocols, such as bendamustine and fludarabine, requiring a couple of months. In cases where a patient’s disease is highly aggressive and the treating team cannot wait for the wash-out and manufacturing period, readily available novel antibodies may offer a viable treatment option. Furthermore, it is important to take into account the patient’s ability to travel when deciding between CAR-T therapy, which is typically a one-time procedure, versus novel antibodies that must be administered regularly until disease progression. An additional crucial factor to consider is the limited accessibility of CAR-T therapy, which is usually available only at a few specialized centers. In contrast, novel antibodies may be more widely available and accessible in a greater number of clinical facilities. However, a significant unresolved issue that remains is the prohibitively high cost of both treatment approaches. Therefore, biomarker identification is crucial to delivering the best treatment to the most suitable patients, ultimately optimizing the use of these expensive therapies. The other question that remains unanswered is which of the novel antibodies is the best. Certainly, factors such as the approval status and the requirement of combining a specific antibody with other agents, which may render some patients ineligible, are important. In addition, comparing the outcome data of novel antibodies against each other is also crucial. So far, there has been no head-to-head comparison. However, a recent comparative analysis that conducted cross-comparisons of several novel antibodies concluded that glofitamab does not appear to confer any advantage in terms of OS when compared to tafasitamab, polatuzumab, and loncastuximab [115].

## 23. Implications for DLBCL Genetic Subtypes

Recent advancements in our understanding of DLBCL have unveiled that the genetic landscape extends beyond the well-established cell of origin and the known genetic alterations involving c-MYC, BCL2, and BCL6. Furthermore, additional four subtypes, namely EZB (21.8%), BN2 (14.8%), MCD (8%), and N1 (2.1%), have been recently identified, collectively accounting for 47% of all cases [116]. Despite these findings, the current treatment approach for LBCL patients remains largely uniform, regardless of their genetic background. However, the emergence of novel antibodies offers a potential opportunity to personalize treatment strategies based on specific LBCL risk and genetic subtypes. In recent studies, such as LOTIS-2, promising responses have been observed with loncastuximab in patients with double-hit, triple-hit, advanced-stage, transformed disease, and refractory DLBCL. These findings suggest that this novel antibody could address a crucial unmet need for these particular patient populations [57]. Comparable results were observed in real-world settings with glofitamab, demonstrating an ORR of 70% in patients with r/r non-GCB DLBCL [117]. By elucidating the genetic characteristics, it may become possible to improve the outcome of patients with high-risk disease and tailor treatment according to the affected molecular pathway. Furthermore, integrating these agents in chemotherapy regimens or combining them with novel small molecules may improve outcomes. For example, in recent preclinical research, venetoclax, a BCL-2 inhibitor, enhanced in vitro antibody-dependent cellular phagocytosis of novel anti-CD20 and anti-CD19 antibodies via macrophages in BCL2-expressing double-hit lymphoma [118].

As discussed above, several ongoing clinical trials aim to determine the effectiveness and safety of several novel treatment regimens in various LBCL risk/genetic subtypes. These advancements offer the potential to shift LBCL treatment towards a more precise and individualized approach based on risk/genetic profiles. This personalized approach holds promise for improving treatment outcomes while minimizing AEs and unnecessary costs.

## 24. Applying Novel Antibodies in Public vs. Private Healthcare Systems

Owing to the substantial expenses associated with novel antibodies, their utilization can differ across public and private healthcare systems. Public healthcare systems often have limited resources and budget constraints, which can impact the availability and accessibility of novel antibodies. On the other hand, private healthcare systems generally offer more flexibility in terms of accessibility to novel therapies. They may have a wider range of reimbursable treatment options and provide quicker access to innovative treatments due to the fewer bureaucratic processes involved and shorter waiting times.

Nonetheless, access to novel antibodies in the private sector may be contingent upon factors such as insurance coverage, out-of-pocket expenses, and the preferences of healthcare providers. In contrast, the public sector typically prioritizes treatments based on cost-effectiveness and often imposes stricter criteria for determining patient eligibility.

However, it is important to note that the availability and accessibility of novel antibodies in both public and private healthcare systems can be influenced by factors such as regulatory approvals, reimbursement policies, and regional variations in healthcare resources. Therefore, the applicability of novel antibodies may ultimately depend on the specific context and healthcare infrastructure of each country.

## 25. Conclusions

We are fortunate to be living in a time of a paradigm shift involving a transition from traditional chemotherapy to more advanced therapies that involve biological agents. These new therapies have the potential to offer several benefits, including fewer side effects, and may potentially expand the pool of eligible patients beyond those who can tolerate chemotherapy, particularly elderly patients who may tolerate novel antibodies. Although our experience with novel antibodies is currently limited, it is expected to increase significantly in the coming years. As we strive to determine the best sequence of therapy for patients with LBCL, we anticipate that these novel antibodies will be introduced earlier in the treatment process than CAR-T due to their favorable side effect profile, less complicated production process, and combinability with other agents. Designing ADCs that can target more than one antigen on lymphoma cells and combining novel antibodies with others, among themselves and with CAR-T, will certainly maximize the benefit. However, it is important to acknowledge that there may be unforeseen side effects associated with such combinations. Therefore, it is imperative we develop biomarkers that can maximize the benefits while minimizing the number of patients who may be susceptible to adverse events but do not derive any benefit. Additionally, new areas of clinical research should be promoted and dedicated to exploring ways to mitigate these adverse events.

## Figures and Tables

**Figure 1 cells-12-01858-f001:**
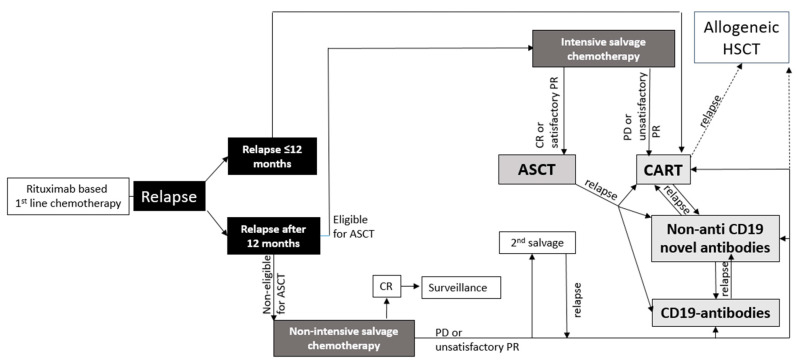
Model for possible treatment algorithm for patients with LBCL.

**Figure 2 cells-12-01858-f002:**
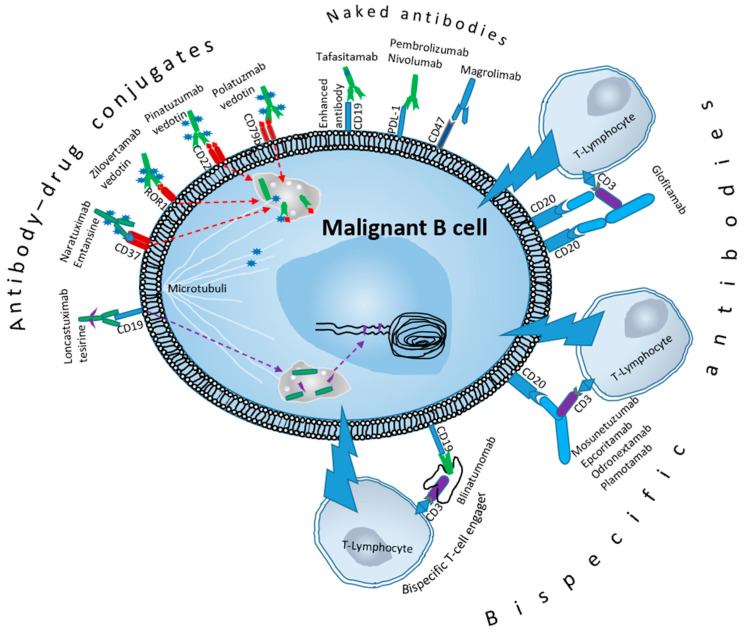
Overview of different novel antibodies and their targets.

**Table 1 cells-12-01858-t001:** Key studies on novel antibodies in patients with r/r LBCL.

Reference	Agent	*n*	Study Phase	Targeted Structure	Combination/Single Agent	CR/PR (%)	PFS (m)	OS (m)
Sehn et al., 2020 [44]	Polatuzumab vedotin	80	Phase 2	CD79b	Pola-BR vs. BR	40 vs. 17.5/0 vs. 0	9.2 vs. 3.7	12.4 vs. 4.7
Motrschhauser et al., 2019 [43]	Polatuzumab vedotin	39	Phase 2	CD79b	Rituximab	21/34	5.6	20.1
Motrschhauser et al., 2019 [43]	Pinatuzumab	42	Phase 2	CD22	Rituximab	26/33	5.4	16.5
Levy et al., 2021 [55]	Naratuximab Emtansine	80	Phase 2	CD37	Rituximab	43.2/32.4	NR	NR
Wang et al., 2021 [56]	Zilovertamab vedotin	13	Phase 1	ROR1	Single agent	23.1/15.4	NR	NR
Hamadani et al., blood 2021 [60]	Loncastuximab tesirine	139	Phase 1	CD19	Single agent	23.4/19	3.1	7.5
Caimi et al., 2021 [57]	Loncastuximab tesirine	145	Phase 2	CD19	Single agent	24.1/24.1	4.9	9.9
Duell et al., 2021 [30]	Tafasitamab	80	Phase 2	CD19	Tafa-Lena	57.5/40	11.6	33.5
Topp et al., 2022 [63]	Glofitamab	46	Phase 1b	CD19:CD3	Glofi-RCHOP	93.5/76.1	NR	NR
Hutchings et al., 2021 [64]	Glofitamab	127	Phase 1	CD19:CD3	Glofi- obinu	48/33.1	NR	NR
Dickinson et al., 2022 [65]	Glofitamab	154	Phase 2	CD19:CD3	Single agent after obinu	52/39	4.9	12-m: 50%
Advani et al. 2018 [40]	Magrolimab	15	Phase 1	CD47	Magro-R	40/33	NR	NR
Maakaron et al. 2022 [41]	Magrolimab	33	Phase 1b	CD47	Magro-RGemOx	51.5/39.4	3.9	12-m: 67.1%
Zinzani et al., 2021 [37]	Pembrolizumab(in PMBCL)	53	Phase 2	Anti-PD-1	Single agent	41.5/20.8	4.3	22.3
Budde et al., 2022 [69]	Mosunetuzumab	129	Phase1/2	CD20:CD3	Single agent	34.9/19.4	7.6	
Olszewski et al., 20222 [70]	Mosunetuzumab	54	Phase1/2	CD20:CD3	Single agent	56/42	NR	NR
Phillips et al., 2020 [71]	Mosunetuzumab	36	Phase1b/2	CD20:CD3	Mosu-RCHOP	96/85	NR	NR
Abrisqueta et al., 2022 [73]	Epcoritamab	29	Phase 1/2	CD20:CD3	Epco- R-DHAX/C	100/80	NR	NR
Phillips et al., 2022 [72]	Epcoritamab	157	Phase 2	CD20:CD3	Single agent	63/39	NR	NR
Bannerji et al., 2022 [74]	Odronextamab	45	Phase 1	CD20:CD3	Single agent	40/35.6	NR	NR
Kim et al., 2022 [75]	Odronextamab	121	Phase 2	CD20:CD3	Single agent	53/37	NR	NR
Bannerji et al., 2022 [74]	Odronextamab	10	Phase 1	CD20:CD3	Single agent	60/60	11.1	NR
Viardot et al., 2016 [78]	Blinatumomab	21	Phase 2	CD19:CD3	Single agent	43/19	3.7	5
Coyle et al., 2020 [79]	Blinatumomab	41	Phase 2	CD19:CD3	Single agent	37/22	2.5	11.2

Abbreviations: *n*: number of patients; m: month; Pola-BR: polatuzumab vedotin with bendamustine and rituximab; Tafa-Lena: tafasitamab–lenalidomide; Glofi-RCHOP: glofitamab plus rituximab, cyclophosphamide, doxorubicin, vincristine, and prednisone; obinu: obinutuzumab; Magro-R: magrolimab–rituximab; RGemOx: rituximab plus gemcitabine and oxaliplatin; Mosu: mosunetuzumab; Epco- R-DHAX/C: epcoritamab, rituximab, dexamethasone, cytarabine, and oxaliplatin or carboplatin. NR: not reported.

**Table 2 cells-12-01858-t002:** Challenges of novel antibodies and future perspectives in management of LBCL.

	Challenge	Possible Future Management
ADCs	Loss of CD79 b expression	BsAbs
	Overexpression of efflux pumps	Drug efflux pump inhibitors
	Activation of compensatory signaling pathways	Novel payloads
	Downregulate immune checkpoint proteins or upregulate inhibitory proteins	Combination with checkpoint inhibitors
	Hypoxia and vascularization	HIF-1α inhibitors
	Neurological AES	Inhibitors of nicotinamide phosphoribosyl-transferase
BsAbs and naked antibodies	Shedding of CD20	Pretreatment with anti-CD20-directed therapy
	Loss of CD20	ADCs
	Hypoxia and vascularization	HIF-1α inhibitors
	P53 mutation	MDM2 inhibitor
	Downregulation of CD3	Combination with checkpoint inhibitors
	ICANS	Lenzilumab
	CRS	Tocilizumab
	Complement activation	Avoid type IgG1 and 3 and use IgG4/IgM

Abbreviations: BsAbs: bispecific antibodies; ADCs: antibody–drug conjugates; HIF-1α: hypoxia-inducible factor-1α; MDM2: mouse-double-minute-2, ICANS: immune-effector-cell-associated neurotoxicity syndrome; CRS: cytokine release syndrome.

## Data Availability

Not applicable.

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
