# Peer review of "Redefining Precision Management of r/r Large B-Cell Lymphoma: Novel Antibodies Take on CART and BMT in the Quest for Future Treatment Strategies"

_cells, 2023, doi:10.3390/cells12141858_

Round 1

Reviewer 1 Report

It is a complete review regarding management of LBCL; it is understandable and clear to the reader first of all because it reports well all the features and efficacy connected to the different "standard approved" therapeutical strategies, including side effects.

Moreover it describes the state of art of new treatments (MOA, phase studies, efficacy, side effects).

Author Response

Thank you.

Reviewer 2 Report

This is an exhaustive review on the current available therapies for LBCL. It covers current treatment strategies, ongoing clinical trials and emerging antibodies.

I would suggest the author to change the brackets on Figure 2.

Are the abbreviations used for antibodies (e.g. Tafasitamab-> tafa) through the text of common use among clinicians?

How are these novel therapies applicable to treatment of the different DLBCL genetic subtypes? Nowadays, all patients receive the same treatment, regardless their DLBCL genetic background. I would advise the author to include a section on this. 

I would also suggest to add a section on how these treatments are applicable in public healthcare system vs. private healthcare systems.

 during 199-2001 Line 40

The English language should be revised, there are minor errors through the text:

e.g. "Beside the in table 1 mentioned novel antibodies (line 666)"

Author Response

I would suggest the author to change the brackets on Figure 2.

Brackets removed.

Are the abbreviations used for antibodies (e.g. Tafasitamab-> tafa) through the text of common use among clinicians?

To avoid any confusion and in agreement with the reviewer, all abbreviations for antibodies were replaced with their corresponding full names. The abbreviated forms were retained in the table to ensure that the flow and size of cells are not disrupted.

How are these novel therapies applicable to treatment of the different DLBCL genetic subtypes? Nowadays, all patients receive the same treatment, regardless their DLBCL genetic background. I would advise the author to include a section on this.

A new paragraph was added discussing this very important topic. Three additional references were subsequently added.

I would also suggest to add a section on how these treatments are applicable in public healthcare system vs. private healthcare systems.

A new section addressing this issue was added.

Some of the English typos/grammar were fixed.

Thank you for your valuable suggestions which certainly will enhance the value of the manuscript.

Reviewer 3 Report

Considering that main indication of discussed drugs is in relapsed/refractory LBCL I suggest to modify the title: "Redefining Precision Management of relapsed/refractory Large B Cell Lymphoma: Novel Antibodies take on CART and BMT in the Quest for Future Treatment Strategies"

Line 68: Does it means a new relapse or refractoriness

line 260: what is "a s a"

No other comments

Author Response

Considering that main indication of discussed drugs is in relapsed/refractory LBCL I suggest to modify the title: "Redefining Precision Management of relapsed/refractory Large B Cell Lymphoma: Novel Antibodies take on CART and BMT in the Quest for Future Treatment Strategies"

Title modified accordingly.

Line 68: Does it means a new relapse or refractoriness

Sentence was corrected.

line 260: what is "a s a"

Sentence was corrected.

I would like to thank the reviewer for the very valuable suggestions which I am happy to follow.